# Cyclic Bond-Slip Behavior of Partially Debonded Tendons for Sustainable Design of Non-Emulative Precast Segmental Bridge Columns

**Leilei Xia [1], Hongcheng Hu [2], Shiyu Guan [2], Yasir Ibrahim Shah [1]** and **Yingqi Liu [1,3,***

[1] School of Transportation and Logistics Engineering, Wuhan University of Technology, Wuhan 430072, China; xialeilei@whut.edu.cn (L.X.); yasiribrahimshah@whut.edu.cn (Y.I.S.)
[2] China Construction Third Bureau First Engineering Co., Ltd., Wuhan 430040, China; huhongcheng2023@163.com (H.H.); guansy91@outlook.com (S.G.)
[3] Department of Civil Engineering, The University of Hong Kong, Hong Kong, China
[*] Correspondence: selbyliu@whut.edu.cn

**Abstract:** The precast segmental bridge columns incorporating resettable sliding joints have been proposed to extend the accelerated bridge construction techniques to regions of moderate to high seismicity while fulfilling the sustainability-based resilient seismic design concept. Following a rethink of the design strategy in the light of inspirations from hybrid sliding-rocking joints, the design of resettable sliding joints can accommodate a certain amount of horizontal sliding displacement and adopt partially debonded tendons in a vertical manner, probably resulting in complicated tensile-flexural loading scenarios in these tendons during earthquakes, which is rarely considered in practice. In this paper, the sustainable design of resettable sliding joints is introduced. A tailor-made setup was established and simplified cyclic bond-slip tests were conducted to validate the practicality of the proposed partially debonded tendon system. Twelve specimens were fabricated using different strands and grouting techniques, and a two-stage numerical model was proposed to interpret the experimental results of seven typical specimens. The results suggest that the deterioration of reloading stiffnesses can be captured by an additional effective length caused by bond failure, and the strands perform mostly elastically under relatively large transverse displacements. The loading stiffness of the anchorage is 26.3 kN/mm, and it has significant effects and the proposed two-stage model can satisfactorily capture the envelope of the response of the partially debonded tendons, providing practical design for the proposed partially debonded tendons used in sustainable non-emulative precast segmental bridge columns.

**Keywords:** bond-slip tests; partially debonded tendons; resettable sliding joints; precast segmental bridge columns

## 1. Introduction

Prefabrication of bridge columns is often preferred because of better efficiency, economy, and quality, as well as less disruption to the site. As shown in Figure 1, 17 pairs of emulative prefabricated bridge columns have been constructed in a local highway renovation project.

However, the application of precast segmental bridge columns (PSBC) is still limited mostly to low-seismicity regions due to the uncertainty over their seismic performance [1]. On the other hand, increasing knowledge of material science and structural dynamics provides all kinds of possibilities for achieving high ductility with minor damage and high energy dissipation capacity. To achieve better sustainability in structural engineering, resilience—i.e., the ability of systems to recover from external disturbance—has therefore been proposed for structural seismic design and has quickly become popular. The

successful application of emulative PSBC in the low-seismicity regions has promoted research on seismic-resistant PSBC with the non-emulative joints, basically, non-grouted dry connections between precast column segments that allow gap opening or relative sliding behavior.

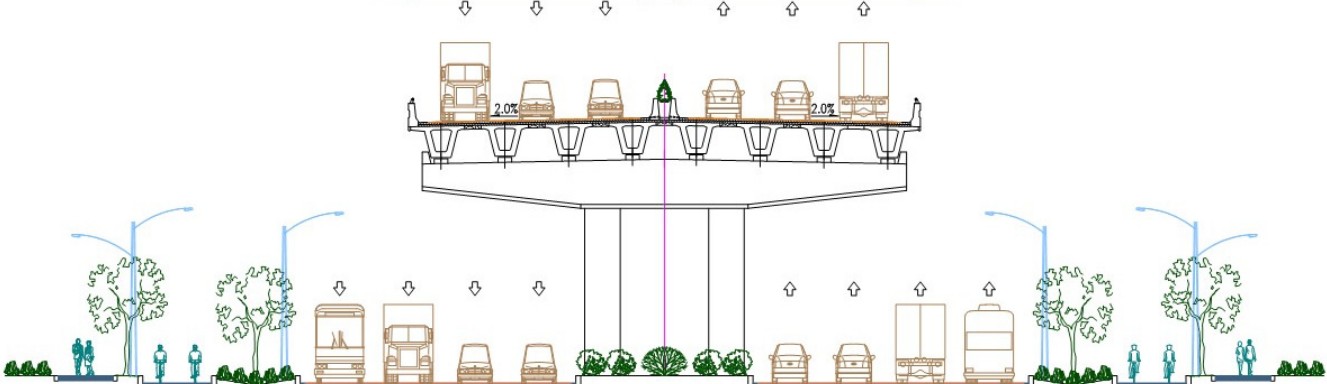

**Figure 1.** Cross section of a city bypass using prefabricated bridge columns.

Earlier non-emulative PSBC were inspired by the discovery of the rocking seismic isolation mechanism. Rocking joints can result in large localized deformation in the vicinity of the opening gaps. Therefore, conventional fully bonded tendons are most likely to rupture before rocking columns reach target drift ratios [2,3]. A debonded region near the rocking joint interface could provide longer effective tendon lengths to distribute the localized deformation, thereby preserving the necessary tendon force for clamping the column segments together and, more importantly, providing some self-centering force for pulling column segments back to their original relative position. A partially debonded tendon system [4,5] and a fully unbonded tendon system [6–10] have been put forward for precast beam-column joints, or PSBC. The subsequent research on non-emulative PSBC with rocking joints has then focused more on the effective use of passive energy dissipation devices at the rocking joints, either in the form of energy dissipation steel bars [6–8,11–16] or as externally replaceable yielding devices [17–21].

With a better understanding of the seismic behavior of PSBC with non-emulative joints, hybrid sliding-rocking (HSR) joints have been proposed using fully unbonded tendons. The drift capacity of the proposed segmental column system with HSR joints in quasi-static pushover tests reaches 15%, and its superior low-damage seismic performance has been observed by shaking table tests [22,23]. Following the similar seismic isolation concept, resettable sliding joints (RSJ) have been proposed [24,25] to further enhance the seismic resilience of PSBC with regard to higher energy dissipation capacity and lower residual drift, where the adopted partially debonded tendon system is new to the industry.

In this paper, the features of the sustainable design of RSJ are introduced. A tailor-made setup was established and simplified cyclic bond-slip tests were conducted to validate the practicality of the proposed partially debonded tendon system. Twelve different specimens were fabricated and a two-stage numerical model was proposed to interpret experimental results. The results suggest that the 7-wire strand performs essentially elastically under orthogonal curvature at 12°, and the anchorage slip helps to dissipate energy. This study indicates that the proposed partially debonded tendon system can be a suitable option for precast segmental bridge columns with RSJs, and proper methods are provided to design the proposed partially debonded tendons used in sustainable non-emulative precast segmental bridge columns.

## 2. PSBC with Resettable Sliding Joints

### 2.1. Overall Design

A conceptual design of PSBC with resettable sliding joints (RSJ) is developed using non-emulative hybrid sliding-rocking (HSR) joints for applications in regions with moderate and high seismicity. In this design, the segments are able to oscillate about the original relative position by sliding and/or rocking for satisfactory seismic isolation during the stage of strong shaking. Moreover, the segments also have the tendency to slide back to the original relative position under the subsequent excitations of smaller magnitudes to achieve satisfactory self-centering performance as if the dislodged segments are reset at the joints upon cessation of an earthquake, thus achieving high resilience and great sustainability toward seismic hazards. In particular, the resettable sliding joint [24] consists of three major components as shown in Figure 2: (i) durable low-friction concrete-to-concrete contact surfaces [26]; (ii) non-planar contact surfaces with gentle guide keys [27]; and (iii) partially debonded tendons with low initial prestress.

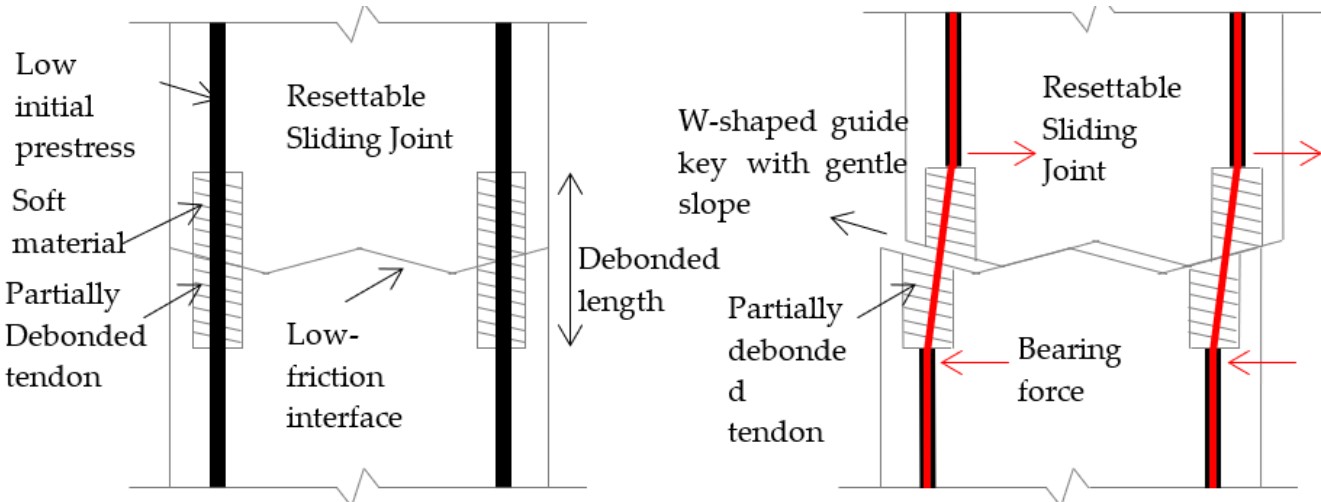

**Figure 2.** Major components of a resettable sliding joint.

### 2.2. Partially Debonded Tendons

Figures 2 and 3 show that the partially debonded tendon comprises a grouted tendon inside a regular duct with a certain length of the duct adjacent to each joint at the contact surface wrapped by an annulus of soft resilient material to allow a certain amount of relative sliding movement at the contact surface during an earthquake. This design not only creates room for relative sliding at the contact surface but also provides some restoring force to assist in resetting. It is, however, necessary to provide sufficient operational prestressing force in the tendons considering the inclination of the gentle guide keys in order to avoid any sliding at joints during conditions of horizontal loading other than earthquakes.

Moreover, unlike the fully unbonded tendons, the presence of grout not only boosts the durability of tendons but also enhances the interaction between adjacent segments. Flexural cracking or crushing of the grout under excessive sliding oscillation during an earthquake is inevitable, but the damage to the grout is considered acceptable during an earthquake.

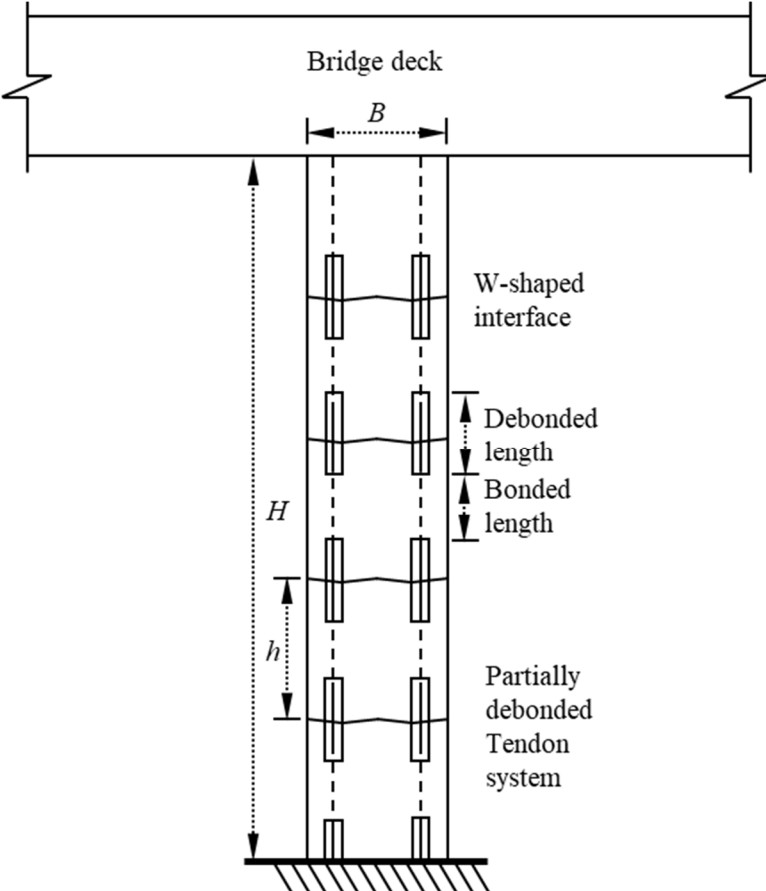

**Figure 3.** Schematic view of precast segmental bridge column with resettable sliding joints.

## 3. Cyclic Bond-Slip Behavior of Partially Debonded Tendons

### 3.1. Bond Behavior of Tendons

According to classical research [28], adhesion, Hoyer's effect, and mechanical interlock are the three controlling mechanisms of the bond between the prestressing strand and concrete. Experiments for investigating the bond performance between the strand and the surrounding materials are usually conducted through pull-out tests [29]. Tests of strands embedded in grout showed that the bond strength increased with a growing positive lateral pressure, embedded length, and grout strength [30]. Other pull-out tests were conducted on strands bonded in the mortar, and the bond strength was higher in specimens with larger strand diameters. Apart from the above obvious factors, the bond force was also reported [31] to be affected by different designs with regard to the concrete cover or spacing between bars and transverse confining steel. Improved bond behavior can be attributed to improved confinement conditions, such as the provision of metal ducts and additional reinforcement. The surface condition of the tendon would also dramatically affect the ultimate bond performance [32].

The primary function of the partially debonded tendon system for the RSJs is to keep the integrity of the assembled column while accommodating some relative joint sliding without causing too much damage. Conventional pull-out tests cannot cater to the complicated tensile-flexural loading scenarios expected during severe sliding conditions. Thus, a simplified cyclic bond test is proposed to investigate cyclic bond performance and energy dissipation of a partially debonded tendon across a sliding joint under vertical sliding displacements.

### 3.2. Overview of Test Setup

#### 3.2.1. Setup and Experiment Design

The Universal Test Machine in the structural lab of the University of Hong Kong is utilized to accommodate specimens with long strands. Figure 4a shows that the machine consists of two major components: a fixed crosshead at the top and a movable loading platform at the bottom. The top crosshead can be adjusted and locked at different levels of the steel-resisting frame. The bottom loading platform can move vertically and is controlled by the hydraulically actuated system. The force capacity is 1000 kN and the vertical stroke of the machine is 273 mm. Displacement-based loading can be applied only in compression (with the bottom platform moving upward). The machine can only be operated manually while providing displacement-controlled axial loading at a slow rate of 1 mm/min to 30 mm/min.

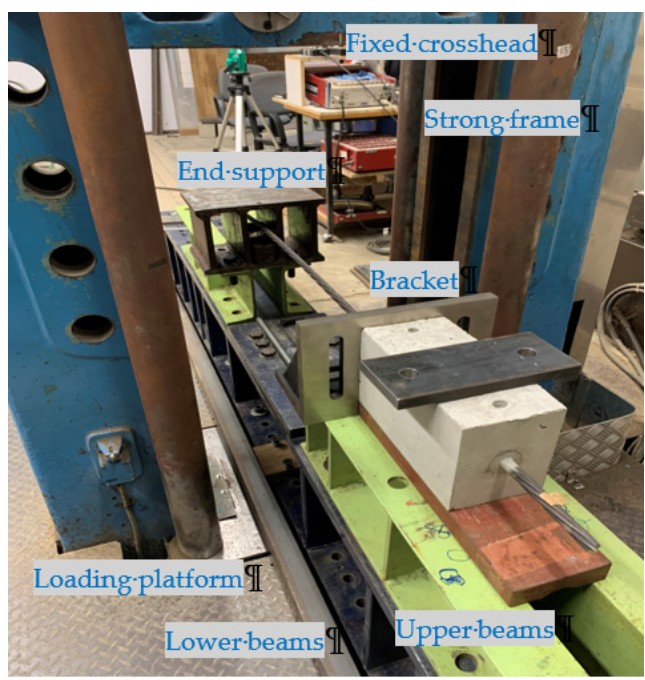

(**a**)

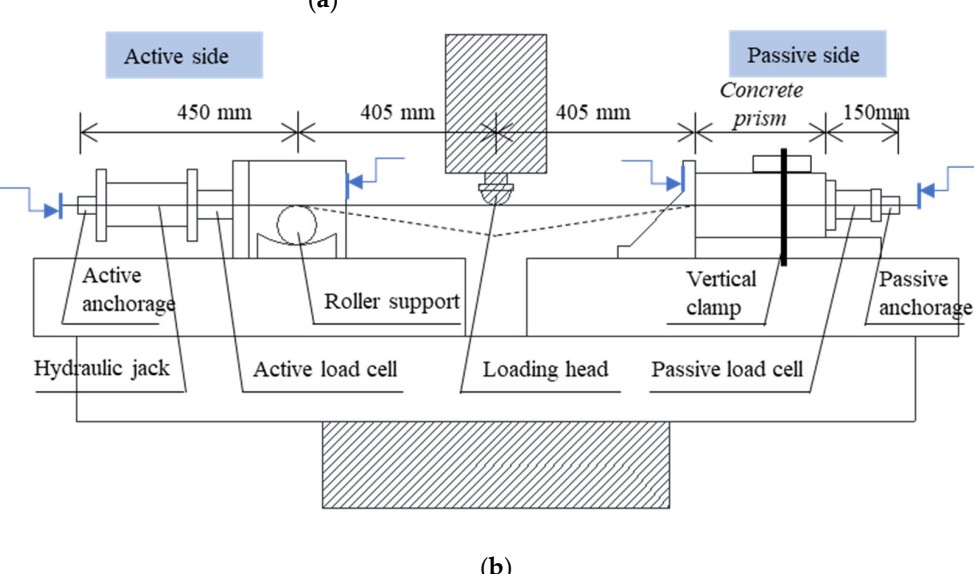

(**b**)

**Figure 4.** Simplified cyclic bond test: (**a**) test setup and (**b**) measurements.

In Figure 4, the specimen, consisting of a concrete prism with a long grouted strand, is mounted horizontally onto the loading platform to simulate a typical partially bonded strand. The unbonded length of the tendon with soft wrapping is simulated by a free length of the strand in the experiment. As shown in Figure 4b, two sets of wedge and barrel anchors were mounted at the two ends of the strand to provide horizontal constraint, while linear variable differential transformers (LVDTs) with 50 mm stoke and load cells with maximum compressive force capacity of 300 kN were installed near these anchors to keep a record of the horizontal movements and forces induced from the imposed vertical midspan displacement. To impose vertical midspan loading displacement onto the specimen, a steel loading head was fixed to the underside of the crosshead and kept stationary throughout the test. The contact surface of the loading head with the strand had a curved shape and its position was adjusted with the help of laser positioning. During testing, the imposed vertical displacement would cause deformation of the strand as depicted by a dotted line in Figure 4b.

At the steel end support, a roller resting on a concave surface was provided. Two LVDTs (with 25 mm stroke) were mounted near the bracket and end support to monitor any incidental horizontal displacements. No prestress was applied to the strand during the fabrication of the specimen. The anchorages were installed manually by hammering the steel wedges into the conical cavity. A hydraulic jack was used to tension the strand before testing. A one-way cyclic testing protocol was adopted as the strand should always be taut during testing.

### 3.2.2. Design of Specimens

Altogether, 12 specimens were cast with grouted strands in two separate batches, and the same C45 concrete mix design was adopted for all the specimens as presented in Table 1. In view of the various factors that may affect the bond performance between the strand and grout, different types of the tendon, grout mix designs, and confining reinforcement are considered.

**Table 1.** Concrete mix design and material properties.

| Batch | Grade | Cement $(kg/m^3)$ | Water $(kg/m^3)$ | Fine Aggregate $(kg/m^3)$ | 10 mm Aggregate $(kg/m^3)$ | 28-Day Cylinder Strength (MPa) | 28-Day Cube Strength (Mpa) |
|---|---|---|---|---|---|---|---|
| First | C45 | 501 | 256 | 857 | 701 | 45.0 | 50.1 |
| Second | C45 | 501 | 256 | 857 | 701 | 44.6 | 51.0 |

Figure 5 shows the concrete prism with a rectangular cross section adopted for the test specimen. Two types of grout were used in the two batches of specimens. Grade 52.5R ordinary Portland cement [33] with a water/cement ratio of 0.4 (which falls in the recommended range of 0.35–0.45 from CEN) was adopted as one option. The other was a commercial product with the brand name of SikaGrout[TM] (Sika, Baar, Switzerland), where a water/Sika-powder ratio of 0.2 was adopted according to the data sheet [34]. Despite the different mix designs, the strength testing results (with 40 mm cube specimens for both grout mixes) indicated that they had similar 7-day compressive strength above 40 MPa, and the ultimate 28-day compressive strength was 60.5 MPa and 65.2 MPa, respectively, for cement grout and Sika grout.

Low-relaxation 7-wire strands of 15.2 mm nominal diameter conforming to BS 5896 [35] were used, and Young's modulus and ultimate tensile strength of these strands at ambient temperature were 200 GPa and 1860 MPa, respectively [36]. The strands used in the first batch were newly bought, and they were coated with oil. In the second batch, strands with dry and clean surfaces were used. The detailed descriptions and labels for different cases are shown in Table 2.

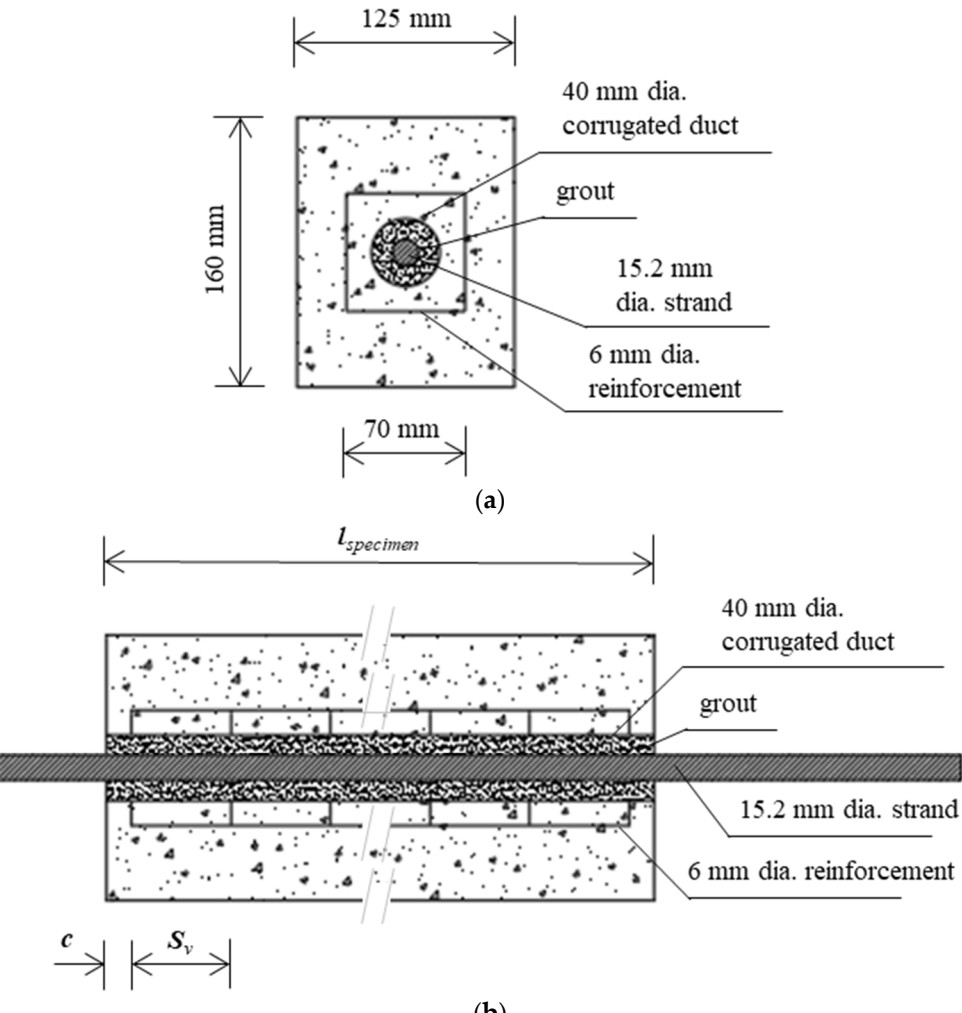

**Figure 5.** Concrete prism in the specimen for simplified cyclic bond test: (**a**) transverse cross section; and (**b**) longitudinal cross section.

Some of the specimens were tested in axial tension only to provide information about the maximum bond resistance as well as to fine-tune the setup and loading protocol for the one-way cyclic bond tests.

### 3.2.3. Fabrication of Specimens

The reinforcement cage was fabricated according to the design and then positioned in the mold. Plastic spacers were used to position the reinforcement cage with cable ties and instant glue to ensure the provision of proper concrete cover. As illustrated in Figure 6, four larger horizontal plastic spacers were used to restrain the reinforcement cage while leaving space for the compaction of concrete.

Care should be exercised during casting to ensure that the fresh concrete could fill the mold completely, including the corners. The four plastic horizontal spacers at the top should be kept until the compaction of the concrete is completed. After 7-day air curing of the freshly cast specimens, 15.2 mm diameter strands were then threaded through the concrete prisms. Any protruding grout lengths of tubes were cut off flush with the surface of the prism. Holes were drilled through the grout tubes and plastic ducts right into the corrugated duct. To avoid leakage of grouting materials, a rubber sheet was attached to the jigs to seal off any gaps. The gaps between the jig and the strand were sealed by plasticine.

**Table 2.** Summary of specimens for cyclic bond tests.

| No. | Batch | Label | Strand Surface Condition | Stirrup | Grout | Length (mm) | Notes |
|---|---|---|---|---|---|---|---|
| 1 | | I-60-270P1 | | 60 | w/c = 0.4 | 270 | axial only |
| 2 | | I-60-270P2 | | 60 | w/c = 0.4 | 270 | axial only |
| 3 | First batch | I-60-270P3 | coated with oil | 60 | w/c = 0.4 | 270 | axial only |
| 4 | | I-60-510P | | 60 | w/c = 0.4 | 510 | axial only |
| 5 | | I-60-270 | | 60 | w/c = 0.4 | 270 | vertical cyclic |
| 6 | | I-60-750 | | 60 | w/c = 0.4 | 750 | vertical cyclic |
| 7 | | II-60-270P | | 60 | w/sika = 0.2 | 270 | axial only |
| 8 | | II-48-270 | | 48 | w/sika = 0.2 | 270 | vertical cyclic |
| 9 | Second batch | II-80-270 | dry and clean | 80 | w/sika = 0.2 | 270 | vertical cyclic |
| 10 | | II-60-270 | | 60 | w/sika = 0.2 | 270 | vertical cyclic |
| 11 | | II-60-510 | | 60 | w/sika = 0.2 | 510 | vertical cyclic |
| 12 | | II-60-750 | | 60 | w/sika = 0.2 | 750 | vertical cyclic |

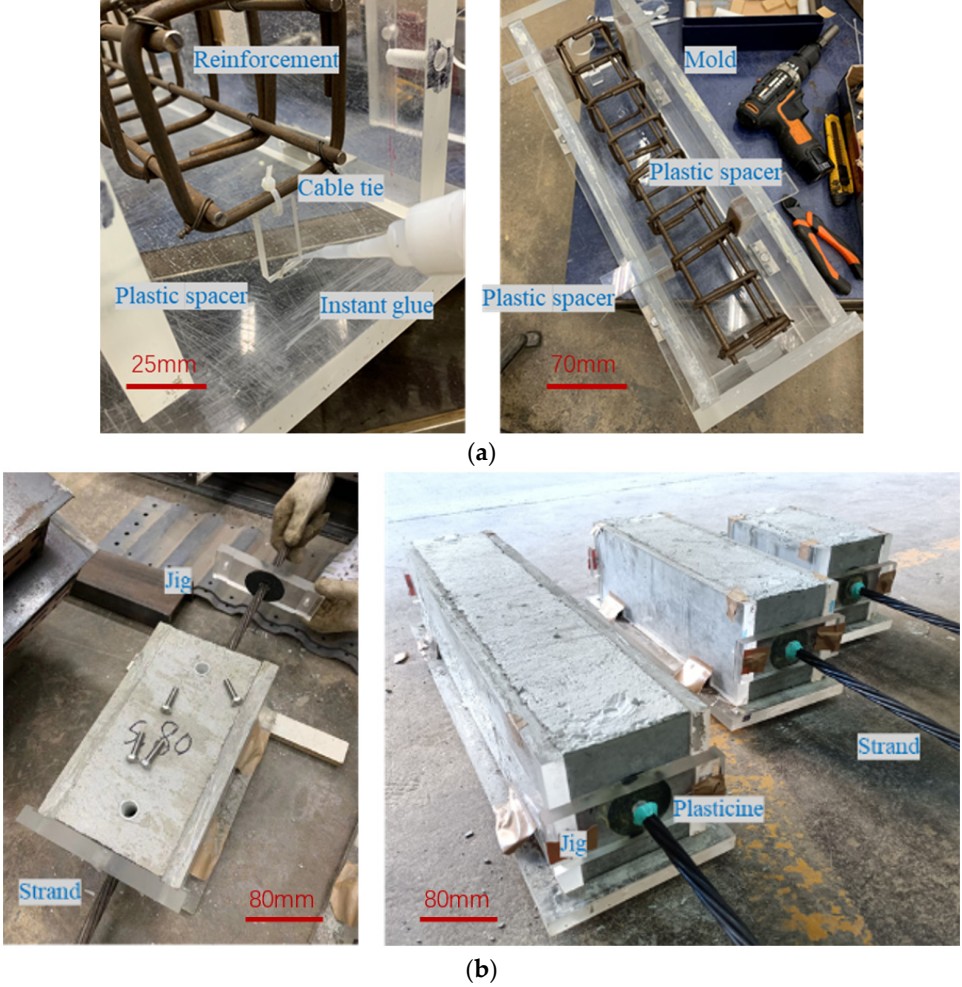

**Figure 6.** Fabrication of the specimen: (**a**) reinforcement positioning; and (**b**) strand threading and fixing at the active side.

Grouting was the last step for specimen fabrication upon completion of strand threading. A tailor-made grout injection tube with a fitted nozzle was used for grouting operations. The specimen was slightly tilted before grouting so that the grout outlet was slightly higher than the grout inlet to allow for the expulsion of air.

### 3.2.4. Testing Procedures

In accordance with the design described in Section 3.2.1, the specimen was mounted onto the testing frame and a very small force was applied by the hydraulic jack to keep the strand taut. Several displacement cycles were imposed while monitoring the force by the active load cell. Upon reaching the target displacement amplitude in a cycle, the specimen would go through force-based unloading to the minimum force (close to 0 kN). The yielding of the stand was not the major concern of the simplified cyclic bond tests because the strand should remain elastic. In the loading or reloading phase of a typical cycle, additional displacement was imposed in accordance with Table 3 by displacement control. Then in the subsequent unloading phase, force control was adopted until the force applied was reduced to zero, possibly resulting in an increase in residual displacement.

**Table 3.** Proposed one-way cyclic loading protocol.

| Stage | Cycles | Loading Rate (mm/min) | Loading and Reloading | Unloading |
|---|---|---|---|---|
| | | | Displacement Control (mm) | Force Control (kN) |
| 1 | | | 5 | to zero |
| 2 | | | 10 | to zero |
| 3 | | 10 | 15 | to zero |
| 4 | | | 20 | to zero |
| 5 | | | 25 | to zero |
| 6 | 2 | | 30 | to zero |
| 7 | | | 40 | to zero |
| 8 | | | 50 | to zero |
| 9 | | 15 | 60 | to zero |
| 10 | | | 70 | to zero |
| 11 | | | 80 | to zero |
| 12 | | | 90 | to zero |

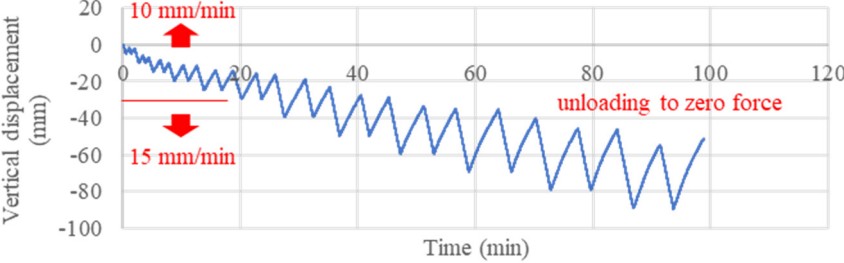

Typical time history for one-way cyclic loading

## 4. Evaluation of Testing Results

### 4.1. Specimens

During the cyclic testing, concentrated vertical loading was applied to the strand by the loading head, causing the strand to deform as indicated in Figure 7. The original roughly round cross section of the 7-wire strand was flattened due to the large vertical force applied from the loading head of the setup, resulting in a permanent curvature. Similar strand damage with smaller permanent curvature was also observed at the roller support and the active side of the concrete prism. As shown in Figure 8, the bonding region located at the passive side of the concrete prism was intact, while the damage observed at the

bonding region on the active side was largely limited to the corrugated duct. No cracking was observed in the concrete prism, and the failure observed was the slipping of the strand.

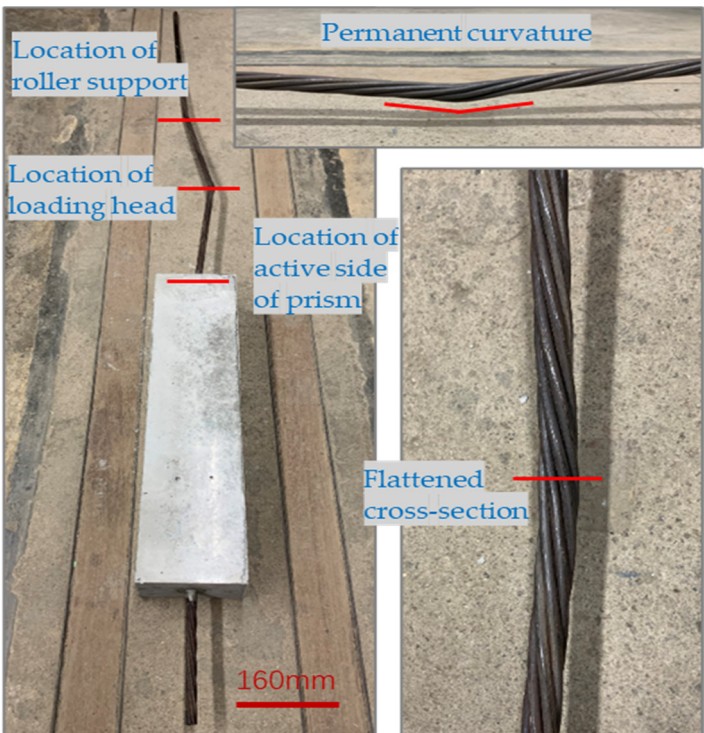

**Figure 7.** Specimen II-60-750 after testing.

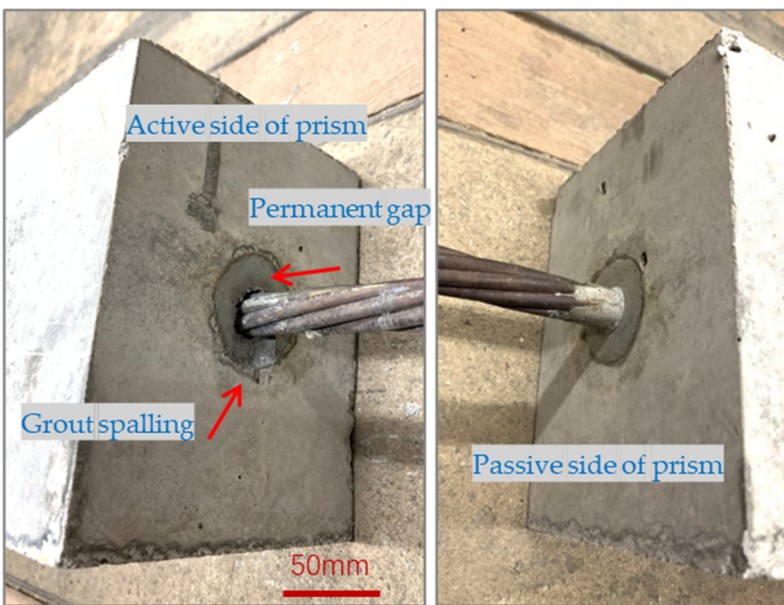

**Figure 8.** Concrete prism of Specimen II-60-750 after testing.

*4.2. Bond Performance*

4.2.1. Bond Parameters

During the axial tensile testing, or the cyclic bond testing, the grout in the vicinity of the strand gradually became detached as the cable force increased. The equivalent bond

stress $\tau_b$ assuming uniform distribution along the embedded length can be calculated from the cable force $P$ acting over the bonded area as shown in Equation (1),

$$\tau_b = P / \left( l_e \cdot l_{specimen} \right) \tag{1}$$

where $l_e$ is the envelope perimeter as shown in Figure 5, which is equal to $(\pi \cdot d_b)$ for a mono-strand tendon, and $l_{specimen}$ is the bonded length as shown in Figure 9.

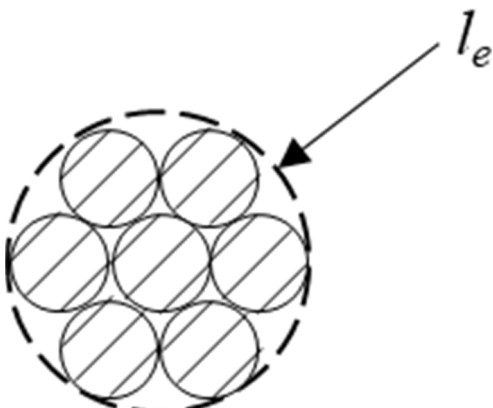

**Figure 9.** Envelope perimeter of the tendon.

As the bond stress may not be evenly distributed, especially for long bonded lengths, the equivalent bond stress $\tau_b$ obtained is indicative only.

### 4.2.2. Cyclic Bond-Slip Performance

During testing, the load cell readings could be used to estimate the bond force in the concrete prism. Similarly, the LVDTs at the prism ends were used to estimate the relative slip of the strand in the concrete prism. The data obtained from the load cells and LVDTs were adopted to investigate the bond-slip performance of the specimens. The typical bond-slip curve for the specimens tested under vertical cyclic imposed displacement is presented in Figure 10.

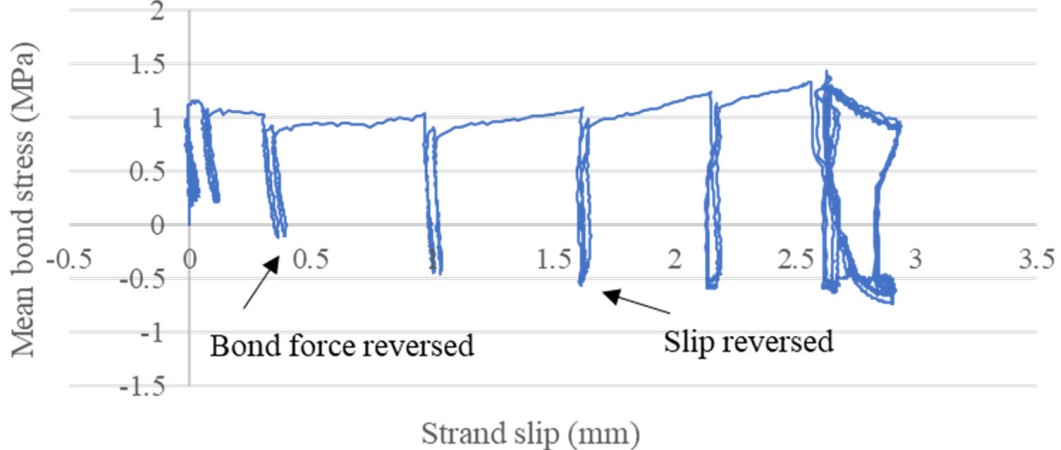

**Figure 10.** The bond-slip curve for Specimen I-60-270.

As the tensile strand force kept increasing, the bond force in the concrete prism would gradually reach its maximum strength and the passive anchorage would start to provide additional resistance together with a positive strand slip toward the active end of the concrete prism. The accumulated resisting force at the passive end of the prism would not be released during the initial stage of unloading, thus leading to a reversed bond force as

the active tensile strand force decreased. A drop in this passive resisting force was only observed with a reversed strand slip.

Owing to the high bond forces in some specimens of the second batch with larger bonded lengths, reversed slip was not observed in Specimen II-60-510 and Specimen II-60-750. For the rest of the cases, the maximum mean bond stresses associated with positive-strand slips were approximately twice the values of the corresponding maximum mean bond stresses obtained during reversed strand slips. The cyclic nature of the imposed displacement might have impaired the residual bond between the grout and the strand. A slight reduction in bond stress was always observed upon unloading and reloading the specimen to the same magnitude of imposed vertical displacement.

### 4.2.3. Mean Bond Stress and Bond Force

Apart from the seven specimens for vertical cyclic tests (Table 3), the other five specimens tested by direct axial tension also provided experimental results on the mean bond stress and the bond force.

The other five specimens in the second batch, as shown in Figure 11, share a fairly stable mean bond stress of about 5.3 MPa. However, the mean bond stresses in all the specimens of the first batch are below 1.2 MPa as shown in Figure 12, indicating that the grouting techniques adopted in the second batch might be capable of providing a higher bond strength between the grout and strand. The oil on the surface of the strands in the specimens of the first batch might have adversely affected the bond performance of grout in the prestressing strand, which was consistent with the findings reported by Borzovič and Laco [32]. From the results of the group of 270 mm long specimens, the adoption of direct axial tension or cyclic testing procedure hardly affected the mean bond stress results.

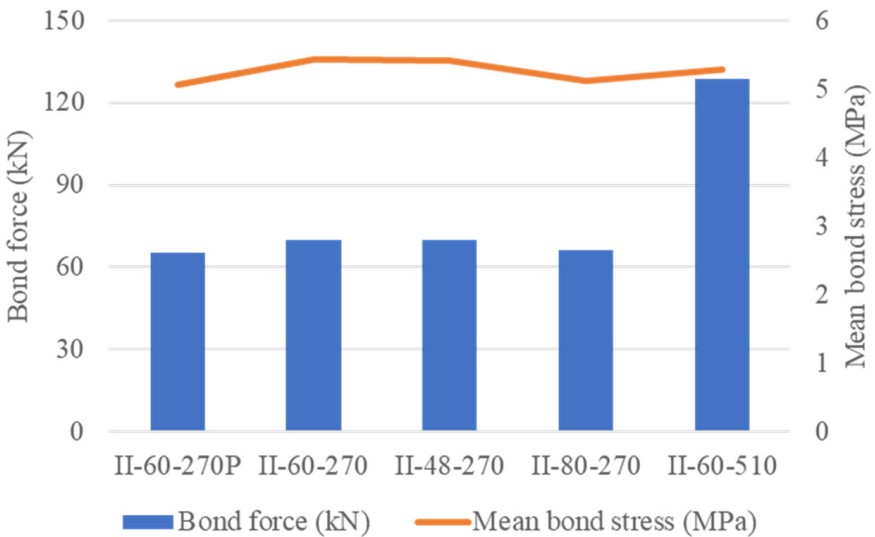

**Figure 11.** Bond performance of specimens in the second batch.

Apparently higher mean bond stresses were observed in specimens with relatively shorter bonded lengths [37–39] as the evenly distributed bond stress assumption was most likely valid. To better understand the relative low bond stresses in first batch specimens, specimen I-60-270P1 was cut through for detailed inspection of the bonding interface between the strand and grout, as shown in Figure 13. Grooves were observed as the strand slid across the bonding interface, indicating that the cement grout failed to keep a tight grip on the strand. This observation could be attributed to the insufficient interlocking effect due to the helical configuration of the strand, the shrinkage of cement grout, and the lubrication effects caused by the oil on the surface of strands in the specimens in the first batch.

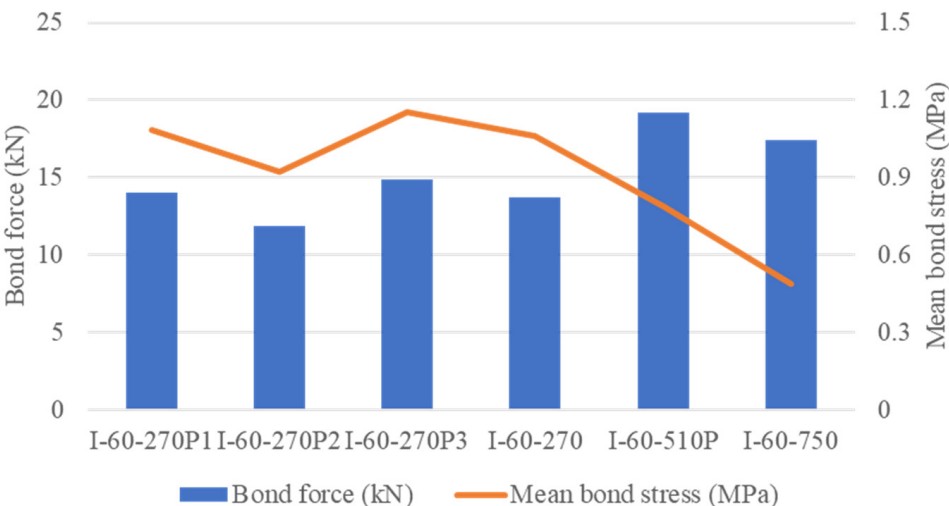

**Figure 12.** Bond performance of specimens in the first batch.

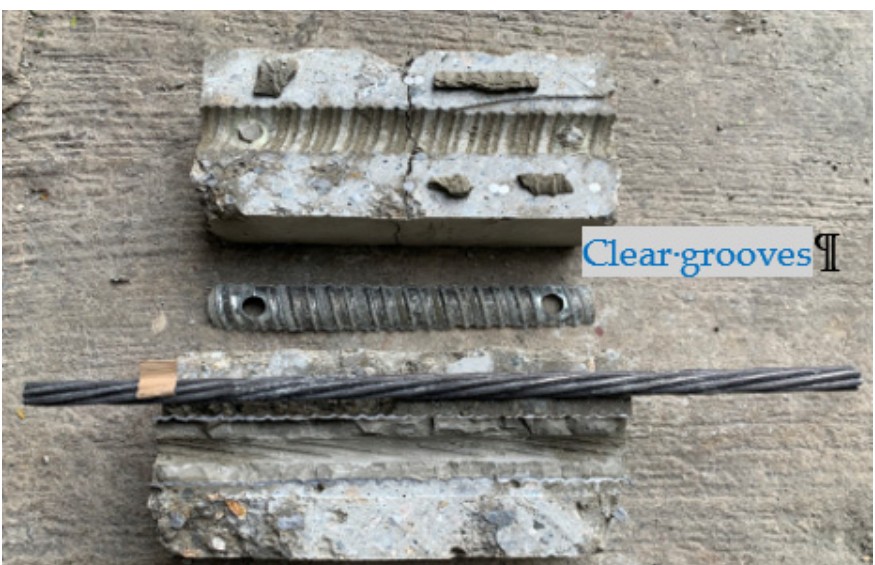

**Figure 13.** Bonding interface between strand and grout in Specimen I-60-270P1.

### 4.3. Cyclic Bond Behaviors

#### 4.3.1. Vertical Forces

The vertical load cell can record the vertical force associated with the V-shaped deformation of the strand. As shown in Figure 14, the relationship between the applied vertical force and the measured horizontal force due to the deformation of the strand during the cyclic testing of Specimen II-60-750 is presented as a representative example. The color of the curve presented gradually changes from black to light gray with the progression of cyclic testing. In each cycle of the imposed vertical displacement, the curve consists of a loading or reloading segment and an unloading segment.

Figure 15 shows that both the vertical force and the horizontal force increase with the progression of testing. A pin-connected truss with simplified geometry, as shown in Figure 15, is used to calculate the vertical force ignoring any bending and shearing effects for comparison.

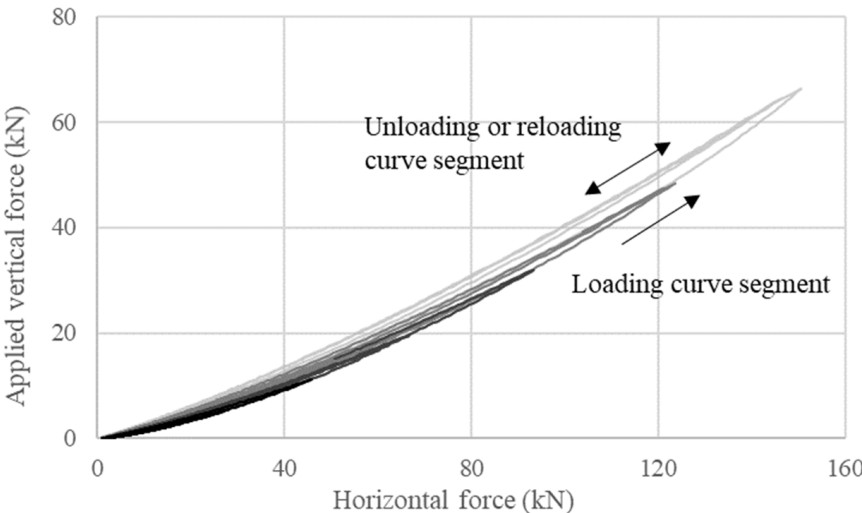

**Figure 14.** Relationship between vertical force and horizontal force during cyclic testing of Specimen II-60-750.

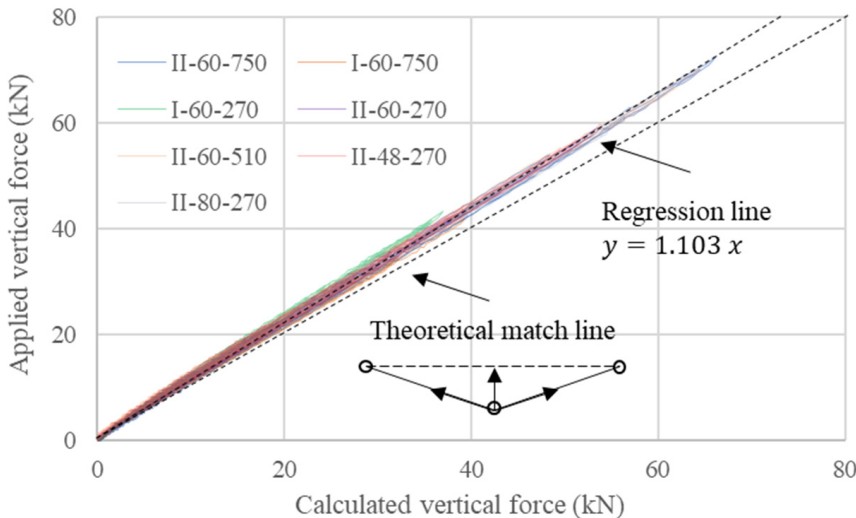

**Figure 15.** Relationship between applied and calculated vertical forces of seven specimens.

4.3.2. Horizontal Forces at Different Vertical Loading Conditions

Figure 16 shows the relationship between the horizontal force measured by the active load cell and the cyclic vertical displacement of Specimen II-60-510 as an example. Similar to Figure 14, each cycle consists of a loading segment and an unloading or reloading segment.

Figure 16 shows that the horizontal force increases rapidly with the progression of testing. In the envelope of loading curve segments, the reduction of the rate of increase is associated with the bond slip. Residual vertical displacement is observed as the measured horizontal force approaches zero upon unloading. This phenomenon is associated with multiple factors, including permanent slip at the anchorage.

The trends of force-displacement histories in other cases and their maximum horizontal forces recorded in the loading curve segments at each vertical displacement are illustrated in Figure 17. Since the results are similar among specimens with different stirrup spacings, Specimens II-48-270, and II-80-270 are excluded. Among the five cases presented, three specimens in the second batch (i.e., II-60-270, II-60-510, and II-60-750) share a similar trend of maximum horizontal force before reaching a vertical displacement of 60 mm. Normally the softening observed is associated with bond-slip failures. Two specimens in the first batch quickly reached bond failures, and their maximum horizontal forces under higher vertical displacements were lower than those obtained in the second batch specimens.

Owing to a longer bonded length and lower bond strength, the lowest maximum horizontal forces were obtained in Specimen I-60-750.

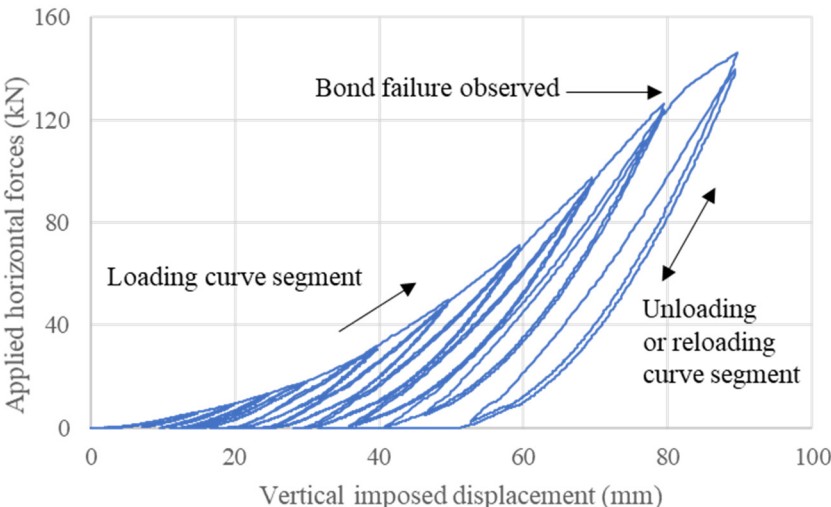

**Figure 16.** Relationship between horizontal force and imposed vertical displacement of Specimen II-60-510.

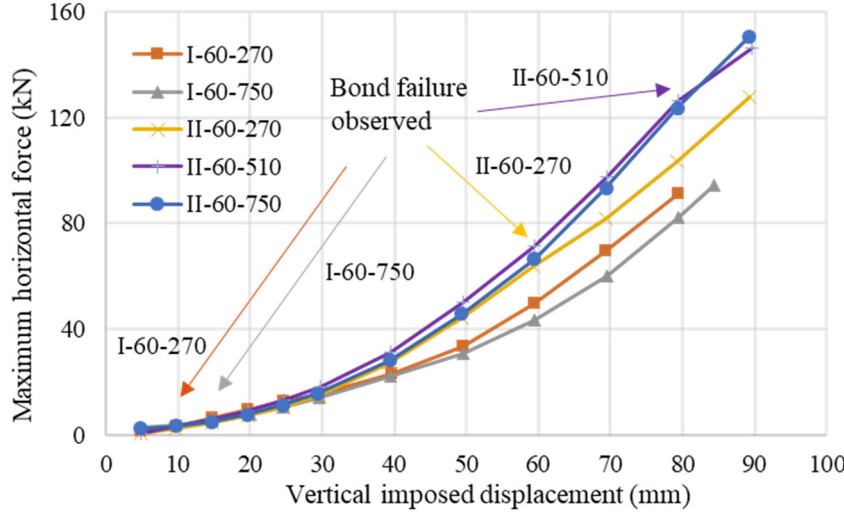

**Figure 17.** Relationship between maximum horizontal force and vertical displacement for selective specimens.

## 5. Further Evaluation of Testing Results

The experimental data in the previous analysis and the observations of cyclic testing suggest that the responses of the specimens are considerably affected by a bond failure in the concrete prism and possible slip at the anchorage(s) at the two ends. To better understand the performance of the proposed partially debonded tendon system, the testing results are further evaluated based on the calculated force-elongation responses of the specimens. A two-stage numerical model is developed and calibrated to describe the envelope of the observed force-elongation responses.

### 5.1. Calculated Axial Elongation

Figure 18 relates the imposed vertical displacement to the simplified triangular geometry, as shown in the dotted line in Figure 18, to derive the axial elongation of the

strand. Considering the movements measured by the active and passive LVDTs, the axial elongation induced by the imposed vertical displacement can be calculated as

$$the \ l_{elongation} = \sqrt{4 \cdot \Delta^2_{vertical} + l^2_{span}} - l_{span} - \Delta_{active} - \Delta_{passive} \qquad (2)$$

where $\Delta_{vertical}$, $\Delta_{active}$ and $\Delta_{passive}$ are the readings recorded in the vertical, active, and passive LVDTs, respectively, and the longer edge of the simplified triangular geometry is $l_{span} = 405 \times 2 = 810$ mm.

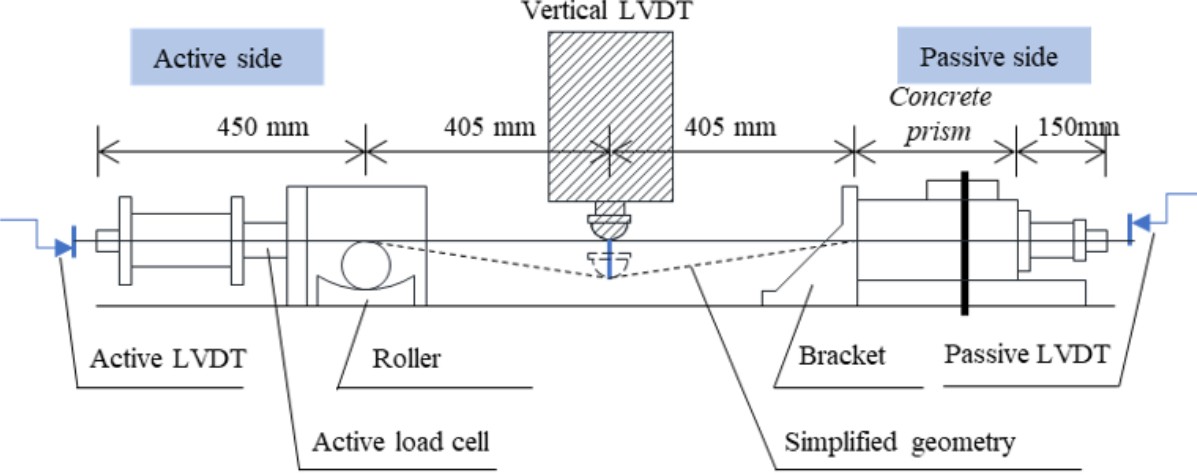

**Figure 18.** Simplified assumptions for calculation of axial elongation.

The experimental results of the two extreme cases are replotted in Figure 19 with the corresponding axial elongation calculated based on the vertical loading conditions. Compared with Specimen I-60-750 with early bond failure, the stiffnesses associated with the loading curve segment of Specimen II-60-750 are much higher, which could be explained by the additional participation of the anchorage at the passive side of the specimen. The early bond failure certainly affected the fixity of the strand in the concrete prism as evidenced by the readings of the passive load cell for Specimen I-60-750.

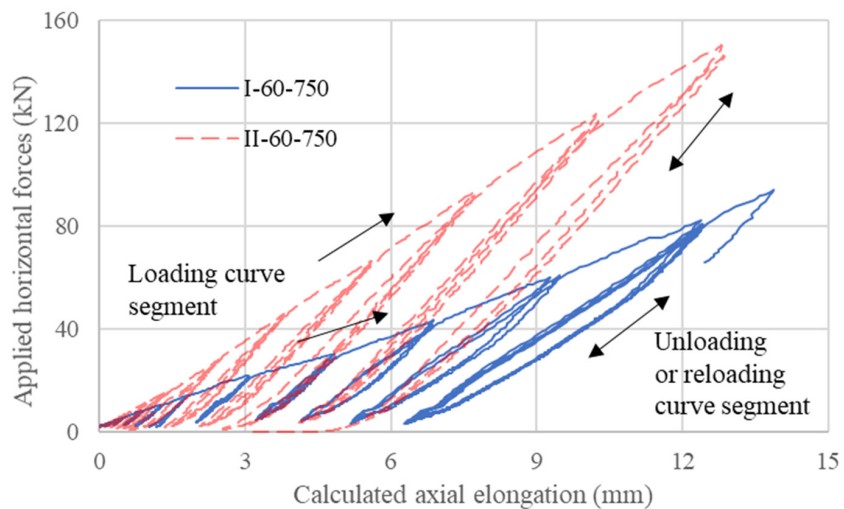

**Figure 19.** Relationship between horizontal force and elongation for two specimens.

### 5.2. Deterioration of Fixity Due to Bond Failure

From the analysis of pull-out testing of strand-anchor systems of Sideris (2012), the wedges slide on the tapered interior surface of the barrel chucks to provide larger radial compression to the strand as the axial load in the strand increases. Owing to the high

friction at the wedge-to-barrel chuck interface, the wedges cannot slide back in the opposite direction as the axial strand force reduces as if the wedges remain "locked" at their ever-reached maximum sliding amplitude. The stiffnesses of the reloading curve segments at different vertical displacements are relevant characteristics. The effective length of the strand can be estimated accordingly.

Figure 20 shows that the original effective length $l_{eff,O}$ of the strand before testing is 1260 mm. The effective elastic stiffness is then calculated based on strand material properties as $k_{eff}$ = 200 GPa × 140 mm$^2$ ÷ 1260 mm = 22.2 kN/mm. According to experimental results presented in Figure 20a, the reloading stiffnesses of specimens II-60-510 and II-60-750 agree well with the effective stiffness, and no deterioration is observed, indicating that the original effective length of the strand is maintained under high loading scenarios. In the other three specimens, as shown in Figure 20b, the reloading stiffnesses deteriorate upon the bond failure. The largest stiffness reduction is observed in Specimen I-60-750, which can be attributed to the lower bond strength and larger bonded length when compared with the other cases as shown in Figure 21. To estimate the additional effective length $l_{eff,add}$ caused by bond failure, one may use

$$l_{eff,add} = \left( \frac{k_{eff}}{k_{trans}} - 1 \right) l_{eff,O} \tag{3}$$

where the $k_{trans}$ denotes the transient reloading stiffness.

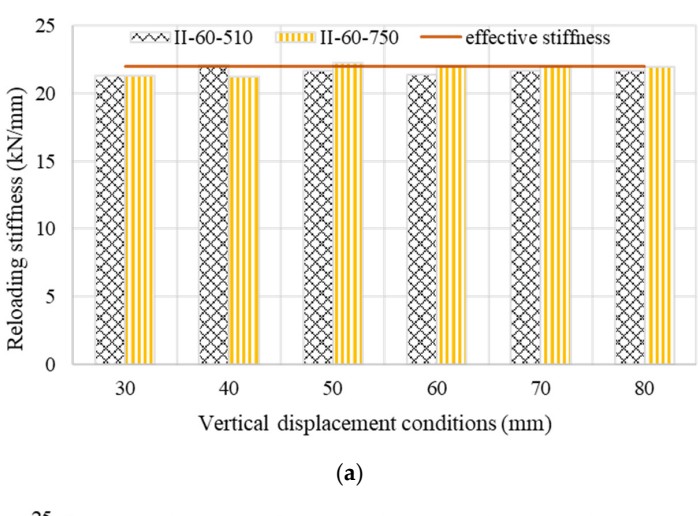

**(a)**

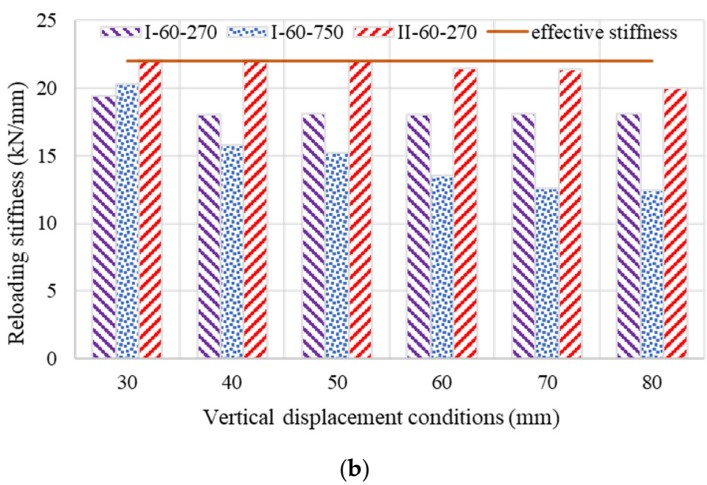

**(b)**

**Figure 20.** Reloading stiffnesses at different vertical displacement conditions for selected specimens (**a**) without deterioration; and (**b**) with deterioration.

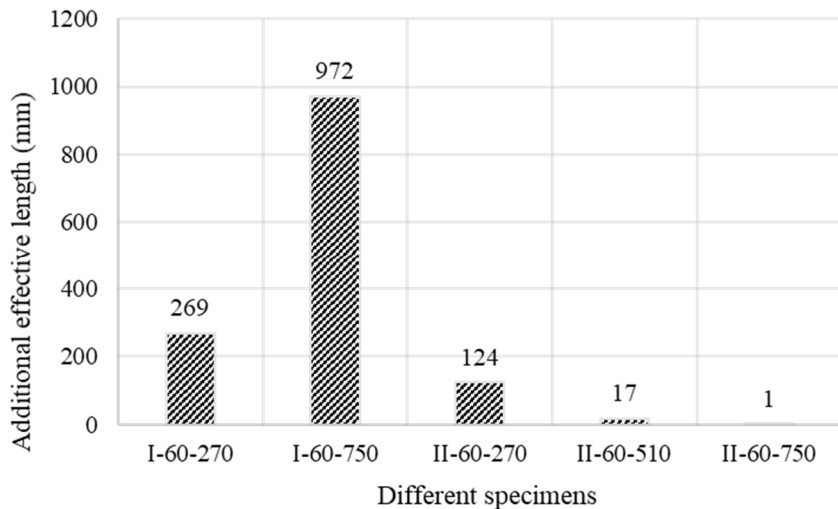

**Figure 21.** Estimation of additional effective lengths at 80 mm imposed vertical displacement.

*5.3. Two-Stage Numerical Model for Partially Debonded Tendon System*

The unloading and reloading curve segments of the horizontal force and elongation histories can be characterized by the effective elastic stiffness $k_{eff}$ and the additional effective length $l_{eff,add}$ calculated by Equation (3) with the transient reloading stiffness $k_{trans}$. However, the loading curve segments of the experimental results in the horizontal force and elongation histories are affected by several factors, including the loading stiffness $k_a$ of the anchorage(s) and the stiffness $k_s$ of the strand.

5.3.1. Establishment of the Model

Referring to the previous model for the direct pull-out strand-anchor system [40] and the observations in the proposed cyclic bond testing, a simplified two-stage numerical model, as shown in Figure 22, is established to describe the envelope of the responses of the partially debonded tendon system.

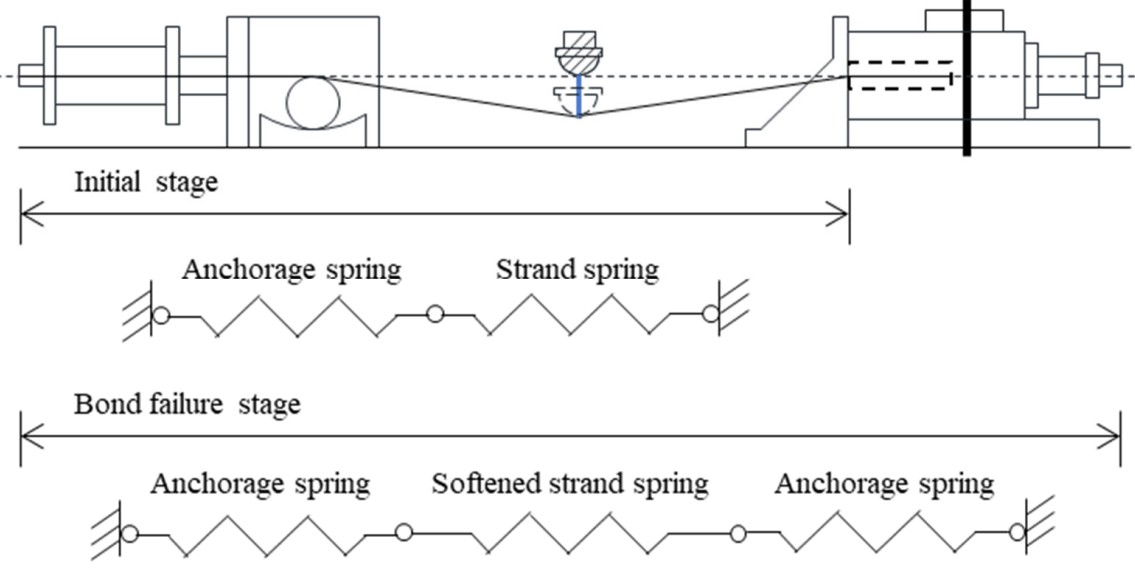

**Figure 22.** A two-stage numerical model for partially debonded tendon system.

Specifically, at the initial stage, an anchorage spring and a strand spring are connected in series to describe the loading curve segment of the partially debonded tendon system. Upon observation of bond failure, the subsequent loading curve segment is described by

three springs in series, where another anchorage spring with identical loading stiffness would be added and the strand spring at the initial stage would be softened by a factor $\gamma$ calculated based on the additional fixity length as

$$\gamma = l_{eff,O} \; / \; \left(l_{eff,add} + l_{eff,O}\right) \tag{4}$$

The force-elongation models adopted for these springs are shown in Figure 23.

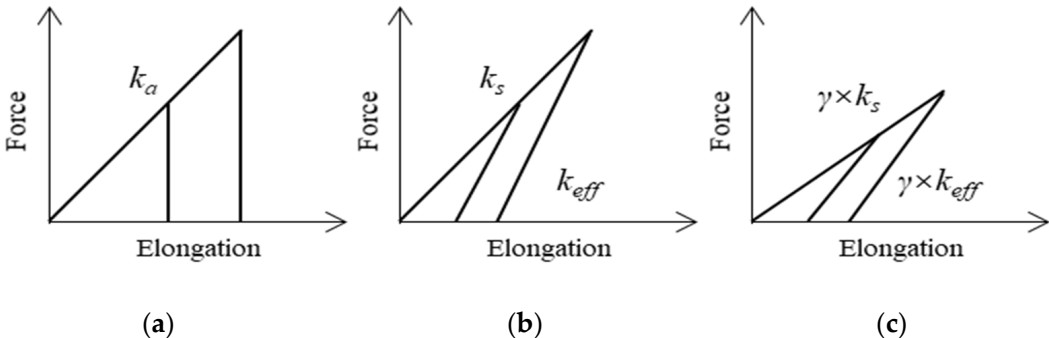

(**a**)             (**b**)             (**c**)

**Figure 23.** Force-elongation models: (**a**) anchorage springs; (**b**) strand spring; and (**c**) softened strand spring.

### 5.3.2. Calibration of the Models

The stiffness of anchorage $k_a$, the stiffness of strand $k_s$, and the additional effective length are calibrated with the representative experimental results of Specimens II-60-270 and II-60-750; the calibrated $k_a$ and $k_s$ are identical while the calibrated additional effective lengths are different in these two cases. The additional effective length is first determined based on the angent value of the unloading and reloading curve segments. Negligible bond damage was observed in Specimen II-60-750, so the additional effective length is zero and there is only one stage in Specimen II-60-750, and the sum of $k_a$ and $k_s$ can be calculated according to the tangent value of the loading curve segment. Considering the additional results in Specimen II-60-270, the $k_a$ and $k_s$ can be determined.

Since $k_s \leq k_{eff}$ and the calibrated $k_s$ is larger than $k_{eff}$, therefore, $k_s = k_{eff} = 22.2$ kN/mm is adopted, and $k_a = 26.3$ kN/mm is determined accordingly. The relevant values are determined from the experimental results to describe the transition between various stages and various loading/unloading curve segments. The corresponding parameters are listed in Table 4, and the related envelopes provided by the calibrated two-stage models are presented in Figures 24 and 25.

**Table 4.** Parameters used for two-stage model calibration.

|  | Unit | II-60-270 | II-60-750 |
|---|---|---|---|
| maximum bond force | kN | 63 | 154 |
| maximum horizontal force | kN | 130 | 154 |
| anchorage stiffness $k_a$ | kN/mm | 26.3 | 26.3 |
| strand stiffness $k_s$ | kN/mm | 22.2 | 22.2 |
| additional effective length | mm | 180 | 0 |
| softening factor $\gamma$ | - | 0.875 | 1 |

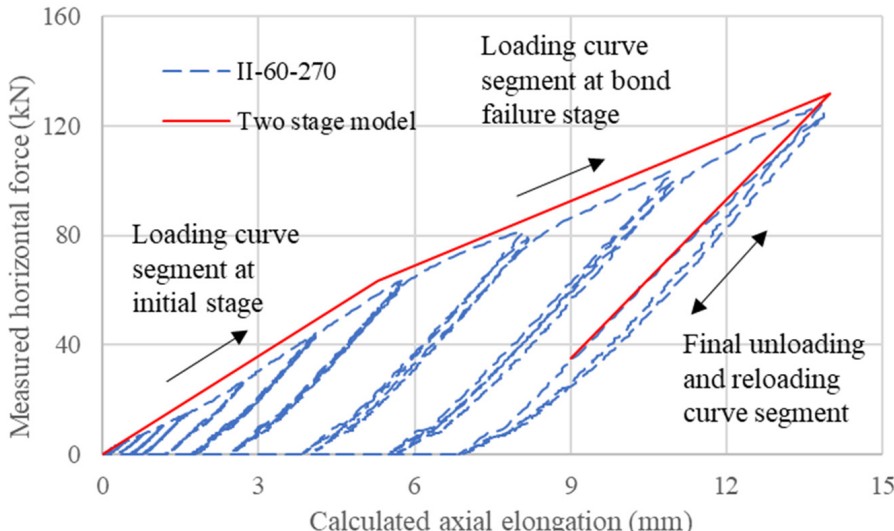

**Figure 24.** Envelope for force-elongation history for Specimen II-60-270.

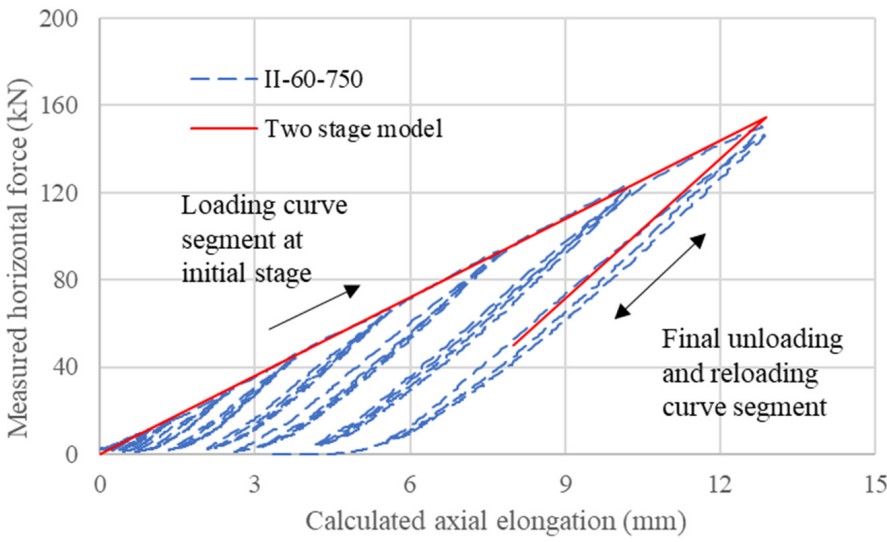

**Figure 25.** Envelope for force-elongation history for Specimen II-60-750.

### 5.3.3. Comparison of Calibrated Model with Other Results

By similar procedures, a comparison between the numerical and experimental results is provided for each of the other three cases and presented in Figures 26 and 27 with the parameters listed in Table 5.

In general, the envelopes provided by the calibrated numerical models are in good agreement with the experimental force and elongation histories in the other three cases, particularly with regard to the value of stiffness at the bond failure stage. For the force and elongation history of Specimen I-60-750, as shown in Figure 28, the stiffness transition between the initial and bond failure stages is unclear, and the additional effective length obtained as 970 mm is much larger than the length of the concrete prism, which can be attributed to the overestimated vertical displacement during the cyclic testing.

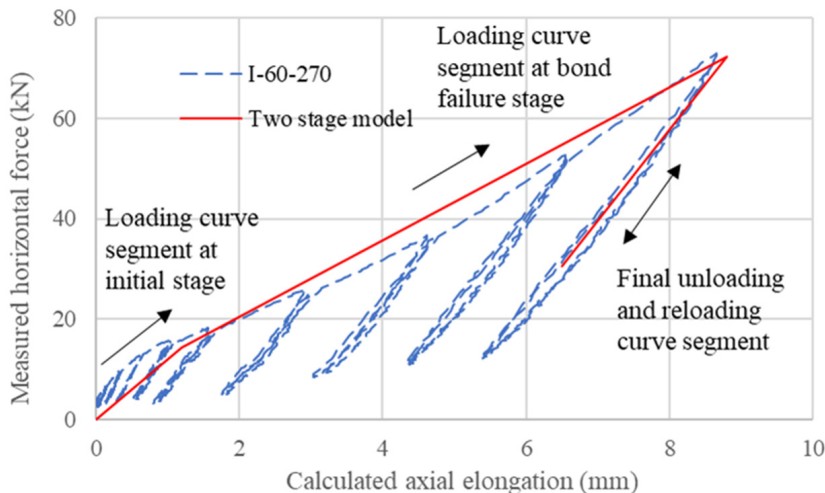

**Figure 26.** Envelope for force-elongation history for Specimen I-60-270.

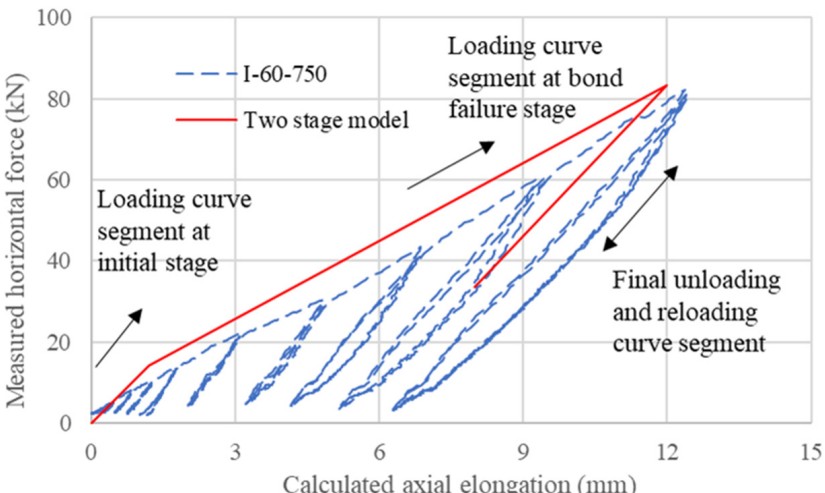

**Figure 27.** Envelope for force-elongation history for Specimen I-60-750.

**Table 5.** Parameters used for comparison.

|  | Unit | I-60-270 | I-60-750 | II-60-510 |
|---|---|---|---|---|
| maximum bond force | kN | 14 | 63 | 126 |
| maximum horizontal force | kN | 73 | 130 | 145 |
| anchorage stiffness $k_a$ | kN/mm | 26.3 | 26.3 | 26.3 |
| strand stiffness $k_s$ | kN/mm | 22.2 | 22.2 | 22.2 |
| additional effective length | mm | 270 | 970 | 70 |
| softening factor $\gamma$ | - | 0.824 | 0.565 | 0.947 |

From the analysis of the response provided by the proposed two-stage numerical model, the contribution of the anchorages to the response of the partially debonded tendon system is significant as the stiffness of the anchorages $k_a$ is comparable to that of the strand. The stiffness of the strand $k_s$ is taken to be equal to the effective elastic stiffness $k_{eff}$, indicating that the strand is performing within its elastic stage.

The anchorage used in this study consists of three steel wedges and a barrel chuck. The calibrated stiffness of the anchorage $k_a$ is equal to 26.3 kN/mm and its unloading/reloading stiffness is taken to be infinite due to the high friction locking effect. However, the choice of such values should be subject to the shape of the anchorage.

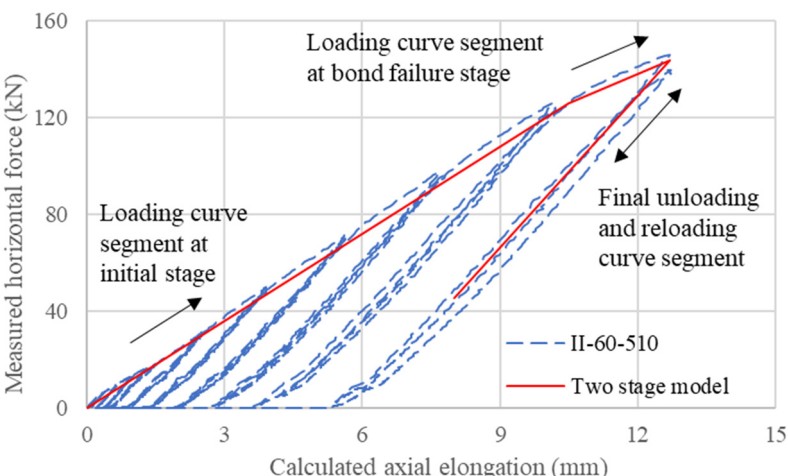

**Figure 28.** Envelope for force-elongation history for Specimen II-60-510.

## 6. Conclusions

In this paper, the sustainable design of PSBC with RSJ is introduced, specimens with grouted strands are prepared to conduct the quasi-static cyclic bond testing, and the response of a typical partially debonded tendon system under orthogonal cyclic displacement is presented and analyzed. A two-stage numerical model is developed to capture the envelope of the observed responses as well as to simulate the contributions from the key components of the partially debonded tendon system. Based on the experimental and simulation results, the following conclusions can be drawn:

- The established setup of the simplified bond testing is suitable to study the cyclic bond behavior of partially debonded tendons. For tendons with relatively low bond stresses, the stirrups in the specimens hardly affect the debonding between the tendon and grout. The direct axial tension and cyclic orthogonal displacement provide comparable values of bond strength.
- The deterioration of reloading stiffnesses can be represented by an additional effective debonded length caused by a bond failure in specimens. For partially debonded tendons with low initial prestress, the gradual fastening of the end anchorages must be considered as the loading stiffness during analysis.
- With proper calibration, a two-stage numerical model with a series of anchorage and strand springs can satisfactorily capture the envelope of the responses of the partially debonded tendon system.
- Despite the observation of permanent transverse deformation after testing, the 7-wire strand performs essentially elastically under large imposed vertical displacement (maximum orthogonal curvature at 12°) in the simplified cyclic bond testing, and the anchorage slip helps to dissipate energy. The proposed partially debonded tendon system can be a suitable option for precast segmental bridge columns with RSJs.

**Author Contributions:** Conceptualization, Y.L.; methodology, Y.L.; software, L.X. and Y.L.; validation, L.X. and Y.L.; formal analysis, L.X. and Y.L.; investigation, L.X. and Y.L.; resources, H.H. and S.G.; data curation, Y.I.S.; writing—original draft preparation, L.X. and Y.L.; writing—review and editing, Y.I.S. and Y.L.; visualization, L.X.; supervision, Y.L.; project administration, Y.L.; funding acquisition, Y.L. All authors have read and agreed to the published version of the manuscript.

**Funding:** This research was partially funded by [the Ministry of Science and Technology, China] grant number [2019YFB1600702].

**Institutional Review Board Statement:** Not applicable.

**Informed Consent Statement:** Not applicable.

**Data Availability Statement:** https://figshare.com/s/f3d763899cd580d5577c.

**Conflicts of Interest:** The authors declare no conflict of interest.

## Abbreviations

| | |
|---|---|
| $c$ | concrete cover to reinforcement in concrete prism |
| $k_a$, $k_s$ | stiffness of anchorage and strand, respectively |
| $k_{eff}$ | effective stiffness of steel mono-strand |
| $k_{trans}$ | transient stiffness of steel mono-strand |
| $l_{eff,add}$ | additional fixity length induced by strand bond failure |
| $l_{eff,O}$ | original fixity length determined by the setup |
| $l_{elongation}$ | axial elongation of strand induced by vertical loading |
| $l_{db}$ | debonded length of partially debonded tendons |
| $l_{span}$ | free length of the strand in cyclic bond testing |
| $l_{specimen}$ | total length of concrete prism |
| $\Delta_{active}$ | measurement recorded in active LVDT |
| $\Delta_{passive}$ | measurement recorded in passive LVDT |
| $\Delta_{vertical}$ | measurement recorded in vertical LVDT |
| $\gamma$ | stiffness softening factor of strand due to bond failures |
| ABC | accelerated bridge construction |
| LVDT | linear variable differential transformer |
| PSBC | precast segmental bridge columns |
| RSJ | resettable sliding joint |

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
