# Peer review of "Cyclic Bond-Slip Behavior of Partially Debonded Tendons for Sustainable Design of Non-Emulative Precast Segmental Bridge Columns"

_sustainability, doi:10.3390/su15108128_

Round 1
Reviewer 1 Report
A very interesting study on prestressed concrete bridge columns design and analysis. From the topic it clearly indicates that a work with good quality results is produced. Secondly the abstract is briefly describing the methodology, scope of the work and results obtained. However, no field application of the proposed work is found in the abstract text's body. It is therefore recommended that a sentence should be added into the text regarding practical applications. Introduction and bibliographic review are fine. Please add a section named significance of current research. Testing scheme, two stage numerical modal introduction and capturing envelope response are all fine. Figures and sketches are precisely drawn illustrating clearly the results and discussion portion of the manuscript. It is pertinent to mention here that at few points some minor grammatical mistakes and errors are found. An english review is therefore recommended.
Quality of English is good. An english review is recommended for correcting minor grammatical and spelling or typo errors.
Reviewer 2 Report
Abstract
There is no justification for the current research.
The abstract could be improved by including quantitative results from the tests.
6 Discussion
The discussion part looks too short. Authors should expand it or include it in Chapter 4.
Authors should read through the manuscript and make the necessary corrections
Reviewer 3 Report
The manuscript entitled "Cyclic Bond-Slip Behavior of Partially Debonded Tendons for Sustainable Design of Non-emulative Precast Segmental Bridge Columns" discusses the development of PSBC in the field of anti-vibration isolation of pier columns in combination with literature, and carries out cyclic bond-slip tests. On this basis, numerical simulation is used to deepen the experimental results. The research logic of the entire manuscript is clear, consistent with the theme of the journal, and can provide better academic and engineering value. The following specific recommendations and opinions are provided for your reference:
1) The manuscript states that the first group of steel strand specimens has slip failure in the early stage, but if consider in vertical disposed placement, the gap between the I-60-750 of tension strand failure and the second group of specimens is only about 5mm. Please explain how the early stage is defined;
2) Section 3.2.4 describes the application of a very small force to the steel strand; if this prestress is achieved by displacement control, the displacement applied or the specific prestress's magnitude should be indicated;
3) In terms of article format, sections 2.1 and 3.1 thoroughly detail the device setup and findings of earlier researchers; it is advisable to discuss this section in the introduction;
4)Section 7 of the discussion part is a little general; it needs to be a detailed discussion and comparative analysis of the research content of the entire manuscript, or improve according to the response of the debonded tendon system under orthogonal cyclic displacement, the inspiration of bond failure in engineering applications, etc.;
5) Can the pictures in the manuscript, such as Figures 3.3, 4.2, 4.3, and 4.7, etc., add some scale labels to the geometric measurements of strands, steel bars, and grout concrete to increase the suggestive nature of the illustrations?
The language expression of the whole manuscript is basically correct, and the logic is clear, but individual expressions and professional vocabulary can be improved slightly.
Reviewer 4 Report
Dear Authors,
Please check the attached file to find the comments.
